

# Antimicrobial activity and synergistic effect of phage-encoded antimicrobial peptides with colistin and outer membrane permeabilizing agents against *Acinetobacter baumannii*

Punnaphat Rothong[1], Udomluk Leungtongkam[1], Supat Khongfak[2], Chanatinat Homkaew[1], Sirorat Samathi[1], Sarunporn Tandhavanant[3], Jatuporn Ngoenkam[1], Apichat Vitta[1,4], Aunchalee Thanwisai[1,4] and Sutthirat Sitthisak[1,5]

[1] Department of Microbiology and Parasitology, Faculty of Medical Science, Naresuan University, Muang, Phitsanulok, Thailand
[2] Office of Disease Prevention and Control Region 3 Nakhon Sawan, Department of Disease Control, Ministry of Public Health, Muang, Nakhon Sawan, Thailand
[3] Department of Microbiology and Immunology, Faculty of Tropical Medicine, Mahidol University, Ratchathewi, Bangkok, Thailand
[4] Center of Excellence for Biodiversity, Faculty of Sciences, Naresuan University, Muang, Phitsanulok, Thailand
[5] Center of Excellence in Medical Biotechnology, Faculty of Medical Science, Naresuan University, Muang, Phitsanulok, Thailand

Corresponding author
Sutthirat Sitthisak,
sutthirats@nu.ac.th

## ABSTRACT

**Background:** *Acinetobacter baumannii* poses a significant public health threat. Phage-encoded antimicrobial peptides (AMPs) have emerged as promising candidates in the battle against antibiotic-resistant *A. baumannii*.

**Methods:** Antimicrobial peptides from the endolysin of *A. baumannii* bacteriophage were designed from bacteriophage vB_AbaM_PhT2 and vB_AbaAut_ChT04. The peptides' minimum inhibitory concentration (MIC) and the synergistic effect of peptides with outer membrane-permeabilizing agents and colistin were determined. Cytotoxicity effects using HepG2 cell lines were evaluated for 24 h with various concentrations of peptides. Biofilm eradication assay was determined using the MIC concentration of each peptide. *Galleria mellonella* infection assay of phage-encoded antimicrobial peptides was investigated and recorded daily for 10 days.

**Results:** The current research indicates that three peptides, specifically PE04-1, PE04-1(NH$_2$), and PE04-2, encoded from the endolysin of vB_AbaAut_ChT04 demonstrated significant antimicrobial activity, with minimum inhibitory concentrations (MIC) ranging from 156.25 to 312.5 μg/ml. The peptides showed antimicrobial activity against multidrug-resistant (MDR) and extensively drug-resistant (XDR) *A. baumannii*, *Escherichia coli*, *Klebsiella pneumoniae*, *Pseudomonas aeruginosa*, *Staphylococcus aureus*, and *Bacillus subtilis*. We found a strong synergistic effect of three peptides with colistin and citric acid, which showed high inhibition percentages (>90%) and low fractional inhibitory concentration (FIC) indexes. The peptides exhibited a high ability to inhibit biofilm formation against twenty *A. baumannii* strains, with PE04-2 showing the most potent

inhibition (91.92%). The cytotoxicity effects of the peptides on human hepatoma cell lines showed that the concentrations at the MIC level did not affect the cell viability. The peptides improved survival rates in the *G. mellonella* model, exceeding 80% by day 10.

**Conclusions/significant finding:** Peptides PE04-1, PE04-1(NH$_2$), and PE04-2 showed sequence similarity to mammalian cathelicidin antimicrobial peptides. They are cationic peptides with a positive charge, exhibiting high hydrophobic ratios and high hydropathy values. The modified PE04-2 was designed by enhancing cationic through amino acid substitutions and shows powerful antibiofilm effects due to its cationic, amphipathic, and hydrophobic properties to destroy biofilm. The peptides improved survival rates in *G. mellonella* infection models and showed no cytotoxicity effect on human cell lines, ensuring their safety for potential therapeutic applications. In conclusion, this study highlights the antimicrobial ability of phage-encoded peptides against multidrug-resistant *A. baumannii*. It can be an innovative tool, paving the way for future research to optimize their clinical application.

## INTRODUCTION

*Acinetobacter baumannii* is a bacterial pathogen commonly found in hospital environments and frequently isolated from medical devices (*Kitti et al., 2023*). It causes severe infections in patients using medical devices, reaching a 48.8% mortality rate (*Kao et al., 2023*). *A. baumannii* has emerged as a significant pathogen due to the increasing prevalence of multidrug-resistant (MDR-AB) and extensively drug-resistant (XDR-AB) *A. baumannii*. Colistin, a cationic lipopeptide, is the last therapeutic option for treating MDR-AB and XDR-AB (*Nović & Jovčić, 2023*). However, resistance to colistin has been reported worldwide. Novel approaches to combat these resistant isolates include phage therapy, which involves using bacteriophage (phage) or phage products for treatment.

Antimicrobial peptides (AMPs) are short (20–30 amino acids), structurally diverse peptides with a broad spectrum of antibacterial activities. They are widely present in nature and are produced by both eukaryotes and bacteria (*Huan et al., 2020*). Phage-encoded AMPs, classified as anti-infective agents, have recently been characterized. Two types of phage-encoded AMPs from endolysin and phage-tail complexes were identified (*Mirski et al., 2019*). Peptides derived from the endolysins of various *A. baumannii* phages have been designed and characterized for their antimicrobial activity (*Thandar et al., 2016*; *Peng et al., 2017*; *Li et al., 2021*; *Lim et al., 2024*).

The action of AMPs is linked to their ability to disrupt microbial membranes, and their cationic charge and hydrophobic properties enable them to interact with and penetrate bacterial membranes, causing cell disruption and death (*Meier et al., 2024*). These cationic peptides, characterized by a net positive charge, show strong affinity for negatively charged

bacterial membranes, leading to membrane disruption, permeabilization, and eventual cell death. Certain cationic peptides, known as cell-penetrating peptides (CPPs), can translocate across biological membranes without disruption, overcoming the impermeable nature of the cell membrane (*Henriques, Melo & Castanho, 2006*).

The synergistic action between outer membrane-permeabilizing agents or antibiotics and AMPs presents a promising strategy to combat antibiotic-resistant bacteria. Outer membrane-permeabilizing agents disorganize and cross the outer membrane, which is crucial in boosting antimicrobial potency, particularly against Gram-negative bacteria (*Farrag et al., 2019*). These agents disrupt the outer membrane (OM), a significant barrier to antibiotic entry, allowing antibiotics to penetrate the bacterial cells and exert their effect. Combining AMPs with antibiotics has permeated bacteria's OM, leading to a synergistic effect (*Gan et al., 2020*; *Zhang et al., 2022*).

While some phage-encoded AMPs have been studied, more research is needed on the role of phage-encoded cationic AMPs. Limited research exists on the interaction between phage-encoded AMPs and antibiotics or membrane-permeabilizing agents. Therefore, this study aims to investigate the antimicrobial activity and potential synergistic effect of phage-encoded AMPs, particularly phage-encoded cationic AMPs, with colistin and outer membrane-permeabilizing agents. Additionally, cytotoxicity and biofilm formation inhibition assays were performed on candidate peptides that demonstrated good antimicrobial activity.

## MATERIALS AND METHODS

### Bacterial strains and growth conditions

Bacterial strains used in this study included *A. baumannii* ATCC19606 and twenty *A. baumannii* clinical isolates collected from tertiary hospitals in Thailand between November 2013 and February 2015 (*Leungtongkam et al., 2018*; *Kongthai et al., 2021*), and from April to June 2022 (this study). Bacterial strains were stored in 25% glycerol stock at −40 °C until use. Bacteria were cultivated on Leed Acinetobacter Agar (LAM) or Luria-Bertani broth (LB) incubated at 37 °C. The use of biological specimens was approved by the Naresaun University Institutional Biosafety Committee (protocol No. NUIBC MI 65-09-34).

### Design of antimicrobial peptides from *A. baumannii* bacteriophage

Two bacteriophages, vB_AbaM_PhT2 (vPhT2) and vB_AbaAut_ChT04 (vChT04), were isolated from hospital wastewater and characterized as previously described (*Kitti et al., 2014*; *Styles et al., 2020*; *Leungtongkam et al., 2023*). The genome sequences of vPhT2 (MN864865) and vChT04 (OQ858591) were obtained from NCBI databases. Antimicrobial peptides consisting of 25–30 amino acid sequences were designed from target proteins of vPhT2 and vChT04. The target proteins of vPhT2 included endolysin (QHJ75684.1), tail lysozyme (QHJ75785.1), short tail fiber (QHJ75776.1), inhibitor of bacterial protein synthesis (QHJ75726.1); and endolysin (WIS40123.1) and tail tubular protein A (WIS40082.1) from vChT04 (Table 1 and Table S1). Active domains of the target

**Table 1 Measured minimum inhibitory concentration (MIC*) values for AMPs from bacteriophage against _A. baumannii_ 19,606.**

| bacteriophage | Phage proteins | Code | MIC values (ug/ml) |
|---|---|---|---|
| vB_AbaM_PhT02; vPhT02 | _ABP02_Endolysin_ | PE02-1 | >2,500 |
| vB_AbaM_PhT02; vPhT02 | _ABP02_Inhibition_ | PIN02 | >5,000 |
| vB_AbaM_PhT02; vPhT02 | _ABP02_Tail lysozyme_ | PTL02 | 5,000 |
| vB_AbaM_PhT02; vPhT02 | _ABP02_Short tail fiber_ | PSTF02 | >5,000 |
| vB_AbaM_PhT02; vPhT02 | _vPhT02(7) modified net charge_ | PE02-2 | 1,250 |
| vB_AbaAut_ChT04; vChT04 | _PCR04_TTPA_ | PTTPAT04 | 2,500 |
| vB_AbaAut_ChT04; vChT04 | _PCR04_Endolysin (hydrolase)_ | PE04-1 | 312.5 |
| vB_AbaAut_ChT04; vChT04 | _vCRTh04-hydrolase-Amides (NH$_2$)_ | PE04-1(NH$_2$) | 156.25 |
| vB_AbaAut_ChT04; vChT04 | _vCRTh04-hydrolase-CPP modified_ | PE04-2 | 156.25 |

**Note:**
  * MIC values were determined with triplicate samples for each peptide concentration using a standard microdilution assay according to CLSI 2020.

protein sequences were analyzed using MOTIF search (https://www.genome.jp/tools/motif/). Antibacterial activity of peptides was predicted using AntiBP2 (_Lata, Mishra & Raghava, 2010_). Peptides were modified and designed using _in silico_ methods to predict efficient cell-penetrating peptides (CPPs) as described by _Gautam et al. (2013, 2015)_. Peptide properties were investigated using the Antimicrobial Peptide Calculator and Predictor (https://aps.unmc.edu/prediction). Antibacterial activity class and subfamily were determined using AntiBP2 (_Lata, Mishra & Raghava, 2010_). The three-dimensional structures of endolysin and peptides were predicted using ChimeraX (https://www.rbvi.ucsf.edu/chimerax/) and AlphaFold2 (_Jumper et al., 2021_). All phage peptides were synthesized by GenScript (New Jersey, USA). Lyophilized peptides were diluted in sterile ultrapure water to prepare 5 mg/ml stock solutions and stored at −20 °C for further use.

## Minimum inhibitory concentration

The MICs of peptides were tested against _A. baumannii_ ATCC19606 and clinical isolates using the broth microdilution method as per the Clinical and Laboratory Standards Institute (_Clinical & Laboratory Standards Institute (CLSI), 2023_). The bacteria were cultured on TSA and incubated at 37 °C overnight. Inocula were prepared by resuspending _A. baumannii_ colonies in 0.85% NaCl solution to a concentration of $1 \times 10^5$ CFU/ml using a densitometer. Two-fold serial dilutions of each peptide were prepared in Mueller-Hinton broth (MHB) in a sterile 96-well U-bottom microtiter plate. Each well was inoculated with 25 μl of bacterial solution, and the plate was incubated at 37 °C overnight. MIC was defined as the lowest peptide concentration that completely inhibited bacterial growth. Peptides with strong antimicrobial activity were further tested against MDR-AB, XDR-AB, and other bacterial species, including _Escherichia coli_, _Klebsiella pneumoniae_, _Pseudomonas aeruginosa_, _Staphylococcus aureus_, and _Bacillus subtilis_ (Table S2).

## Synergism of peptides with colistin and outer membrane-permeabilizing agents

The synergistic effects of peptides with outer membrane-permeabilizing agents (benzoic acid, malic acid, lactic acid, citric acid), EDTA, and colistin were evaluated as described previously (*Sitthisak et al., 2023*). MICs of all agents were assessed, and 0.25× of the MIC of two agents was used in growth inhibition assays (Table S3). *A. baumannii* ATCC19606 was used as the host strain, and inocula were prepared by resuspending *A. baumannii* colonies in 0.85% NaCl solution to a concentration of $1 \times 10^5$ CFU/ml. A total of 50 µl of double-strength Mueller-Hinton broth and 25 µl of bacterial inocula were added to each well of 96-well microtiter plates. Diluted peptides and test agents were added, and the plates were incubated at 37 °C overnight. After incubation, 25 µl of 0.1% triphenyl tetrazolium chloride (TTC) was added to each well and incubated for an additional hour. Absorbance was measured at 540 nm using a microplate reader. The percent inhibition was calculated using the following formula:

$$\% \text{ inhibition} = \frac{\text{the absorbance of controls} - \text{the absorbance of treated wells} \times 100}{\text{the absorbance of control}}$$

Checkerboard synergy testing was performed using peptides and test agents in 96-well plates, varying peptide concentrations across rows and tested agents across columns. *A. baumannii* ATCC19606 inocula ($5 \times 10^5$ CFU/ml) were added, and the plates were incubated overnight at 37 °C. After incubation, 25 µl of 0.1% TTC was added and incubated for an additional hour. The experiment was conducted in triplicate, and results were averaged. The Fractional Inhibitory Concentration Index (FICI) was calculated to determine synergy, with FICI ≤ 0.5 considered synergistic, FICI = 0.5–4 additive or indifferent, and FICI > 4 antagonistic (*Lorian, 2005*).

## The MTT (3-(4,5-dimethylthiazol-2-yl)-2,5-diphenyltetrazolium bromide) assay

We aimed to determine the effects of peptides on cell viability and proliferation in the human hepatocellular carcinoma HepG2 cell line. The MTT assay was performed as described by *Muangpat et al. (2022)* with some modifications. The cells were cultured in Dulbecco's Modified Eagle Medium (DMEM; Sigma-Aldrich, St. Louis, MO, USA), supplemented with 10% fetal bovine serum (FBS), 100 U/ml penicillin, and 100 µg/ml streptomycin at 37 °C in 5% $CO^2$ and 95% humidity. The cells were seeded into a 96-well plate at a density of $1 \times 10^4$ cells/well and allowed to adhere overnight. Then, the cells were exposed for 24 h to various concentrations of peptides. The control cells were cultured in complete DMEM medium containing 0.2% DMSO. After the 24-h incubation period, 20 µL of MTT solution (5 mg/mL in PBS) (Tokyo Chemical Industry Co., Ltd., Tokyo, Japan) was added to each well and incubated for another 2 h at 37 °C. The culture medium was then removed, and formazan crystals formed by viable cells were dissolved in 100 µL of DMSO before measuring the absorbance at 590 nm using a microplate spectrophotometer. The effect of the peptide on cell viability was calculated as a percentage

of the control, which was arbitrarily assigned a value of 100% viability. The percentage of cell viability was calculated relative to the control group. The $IC_{50}$ of the peptides was defined as the concentration that caused a 50% reduction in cell viability compared with the control. The MTT assay was conducted in three independent experiments, each with triplicate wells per condition.

## Inhibition of biofilm formation

We determined the ability of peptides to inhibit biofilm formation in twenty biofilm-forming strains using the method described by *Karyne et al. (2020)* with some modifications. Briefly, biofilm-forming strains of *A. baumannii* were adjusted to an equivalent of the 0.5 McFarland standard in 2× Luria Bertani (LB) broth supplemented with 20% glucose. Then, 50 µL of *A. baumannii* and 50 µL of peptides were added. The MIC concentration of each peptide (PE04-1, PE04-1(NH$_2$), PE04-2) was prepared to final concentrations of 312.5, 156.25, and 156.25 µg/mL, respectively. The plates were incubated overnight at 37 °C. After 24 h of incubation, the culture medium was gently removed, and the wells were washed three times with phosphate-buffered saline. The adherent cells were fixed with absolute methanol for 10 min, stained with 0.4% crystal violet for 15 min, and washed three times with sterile distilled water, then air-dried. The plates were then filled with 250 µL of 33% acetic acid for 15 min. The absorbance at OD595 nm was determined. All experiments were performed in three independent assays, each repeated in triplicate, and the percentage of biofilm reduction was calculated based on treated wells *vs.* non-treated wells.

## *Galleria mellonella* infection assay

The *G. mellonella* infection assay was performed as described by *Peleg et al. (2009)* with some modifications. Final-instar larvae weighing 180–350 mg were used. The antimicrobial activity of peptides against *A. baumannii* ATCC19606 and XDR-AB (AB329) was evaluated on larvae infected with 10 µL of test agents. The peptides (PE04-1, PE04-1(NH$_2$), PE04-2) were tested at final concentrations of 312.5, 156.25, and 156.25 µg/mL, respectively. Each peptide was prepared at 2× MIC concentration and mixed with an equal volume of $1 \times 10^8$ CFU/mL bacteria. Uninfected larvae injected with 10 µL of phosphate-buffered saline (PBS) were used as control groups. Ten worms were used per group. All experimental groups were performed in triplicate. After injection, larvae were incubated at 37 °C in darkness under a humidified atmosphere with food for 10 days. Survival was recorded daily for 10 days. All experimental protocols were approved by the Naresuan University Animal Care and Use Committee (protocol No. NU-AI660602).

## Statistical analyses

All experiments were independently performed in triplicate, and the data are presented as mean ± standard deviation. One-way analysis of variance (ANOVA) with Tukey's comparison test was used to assess statistically significant differences among the experimental groups. A *p* value of $< 0.05$ was considered statistically significant. Survival

data from the *Galleria mellonella* assays were analyzed using Kaplan-Meier survival curves, and differences between treatment groups were assessed using the log-rank test.

## RESULTS

### Characterization of the phage-encoded antimicrobial peptides

We designed ten peptides from endolysin, transcriptional inhibitor protein, tail lysozyme, and tail tubular protein A of bacteriophage vB_AbaM_PhT2 (vPhT2) (MN864865) and vB_AbaAut_ChT04 (OQ858591). The sequences and properties of all peptides are present in Table S1. We modified peptides called cell-penetrating peptides (CPPs) using amino acid replacement. PE04-2 was derived from peptide PE04-1 by amino acid substitutions (Lysine-K) at the N-terminal of peptide PE04-1 to enhance the net charge (Table S1). Additionally, an amine ($NH_2$) was added at the C-terminal of peptide PE04-1 to generate peptide PE04-1($NH_2$). All peptides' molecular weights, hydrophobic ratios, net charges, hydropathy values, and toxin predictions are shown in Table S1. The AMP scores of all peptides ranged from −0.185 to 1.765 (Table S1). The hydrophobic ratios of all peptides ranged from 24% to 52% (Table S1). The hydropathy values of all peptides ranged from −1.57 to 0.38 (Table S1). The sequences and predicted three-dimensional structures of PE04-1, PE04-2, and PE04-1($NH_2$) peptides demonstrated in Fig. 1, showed that PE04-1 and PE04-2 formed the α-helix structure.

### Minimum inhibitory concentration of phage-encoded peptides

Ten peptides derived from bacteriophage proteins were tested for antimicrobial activity. The results showed a wide range of MIC values, indicating varying levels of antimicrobial efficacy. We found three peptides, PE04-1, PE04-2, and PE04-1($NH_2$), showed good activity (MIC ranging from 156.25–312.5 µg/ml), while the MICs of other peptides ranged from 1,250 to more than 5,000 µg/ml (Table 1). PE04-1, PE04-2, and PE04-1($NH_2$) peptides were selected and tested against MDR-AB, XDR-AB, and other bacterial species. PE04-2 demonstrated maximum antimicrobial activity of all tested AMPs, particularly against *E. coli, K. pneumoniae*, and *P. aeruginosa*. The MICs of PE04-2 against MDR-AB and XDR-AB ranged from 78.12 to 312.5 µg/ml (Table S2). PE04-1 was less active against all Gram-negative bacteria tested in this study, while PE04-1($NH_2$) was less active only in *P. aeruginosa*. However, all three peptides showed good antimicrobial activity against *S. aureus* and *B. subtilis*, with MIC values ranging from 19.53 to 39.06 µg/ml (Table S2). Therefore, PE04-1, PE04-2, and PE04-1($NH_2$) were selected for further study.

### Synergism of phage-encoded peptides with colistin and outer membrane permeabilizing agents

We performed an inhibition assay of three peptides combined with colistin, EDTA, and four weak acids (benzoic acid, malic acid, lactic acid, and citric acid) at 0.25× the MIC of two agents. The MICs of all tested agents are shown in Table S3. As shown in Fig. 2, all three peptides exhibited elevated antibacterial activity with all tested agents. The combination of peptide PE04-1 plus colistin, citric acid, EDTA, malic acid, lactic acid, and benzoic acid showed percent inhibition as 99.6%, 99.5%, 72.8%, 70.9%, 66.3%, and 40.9%,

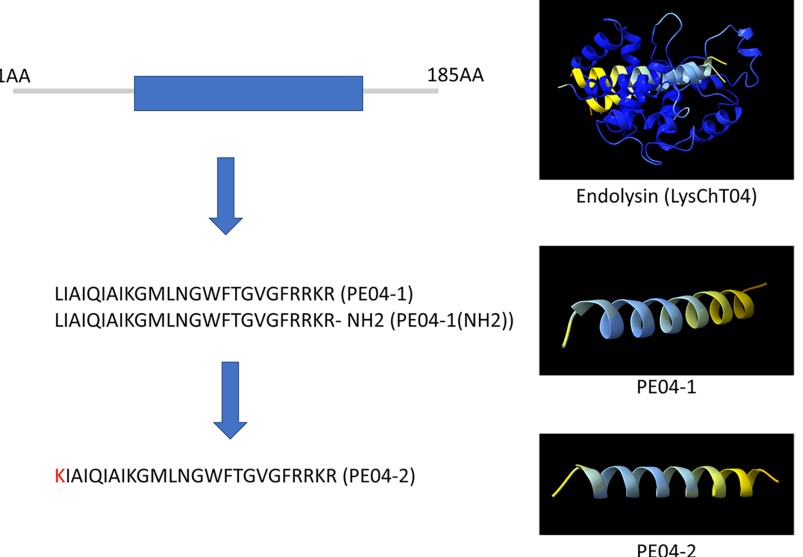

**Figure 1 Schematic diagram of endolysin and phage-encoded antimicrobial peptides from vB_AbaAut_ChT04.** The conserved domain of the putative endolysin from vB_AbaAut_ChT04 was predicted using the Pfam webserver. The three-dimensional structures of endolysin and peptides PE04-1, PE04-2, and PE04-1 (NH$_2$) were predicted using ChimeraX and AlphaFold2.

respectively (Fig. 2, Table S4). The combination of peptide PE04-1(NH$_2$) plus colistin, citric acid, malic acid, EDTA, lactic acid, and benzoic acid showed percent inhibition as 99.9%, 99.8%, 89.9%, 82.79%, 50.3%, 40.1% (Fig. 2, Table S4). The combination of peptide PE04-2 plus citric acid, colistin, EDTA, lactic acid, malic acid, and benzoic acid showed percent inhibition as 99.8%, 99.6%, 99.1%, 86.4%, 85.6%, and 48.7% (Fig. 2, Table S4). We performed a checkerboard assay to measure peptide synergistic effect with colistin and citric acid, which showed a high synergistic effect. The FIC indexes of PE04-1, PE04-1 (NH$_2$), and PE04-2 plus colistin were calculated as 0.19, 0.19, and 0.14, respectively, which indicates synergism. The FIC indexes of PE04-1, PE04-1(NH$_2$), and PE04-2 plus citric acid were calculated as 0.37, 0.37, and 0.25, respectively, indicating synergism (Table S5).

## Cytotoxicity effects of phage-encoded peptides to human hepatoma cell lines

The cytotoxicity effect of three peptides was determined using an MTT assay. All three peptides at 600 ug/ml concentrations exhibited cytotoxicity against the human liver cancer cell line (HepG2). As shown in Figs. 3A–3C, all peptides caused the death of HepG2 cell lines in a dose-dependent manner. The percentage of cell viability of the PE04-1, PE04-1 (NH$_2$), and PE04-2 at the MIC concentration was more than 85% (Figs. 3A–3C). The concentration of the PE04-1 and PE04-1(NH$_2$) that caused the reduction of viable cells to 50% (IC50) was 1,440 and 525.6 ug/ml, while the IC50 of peptide PE04-2 was 424.1 ug/ml (Figs. 3A–3C). From the IC50 values, it could be suggested that concentration at MICs did not affect the cell viability of human hepatoma cell lines.

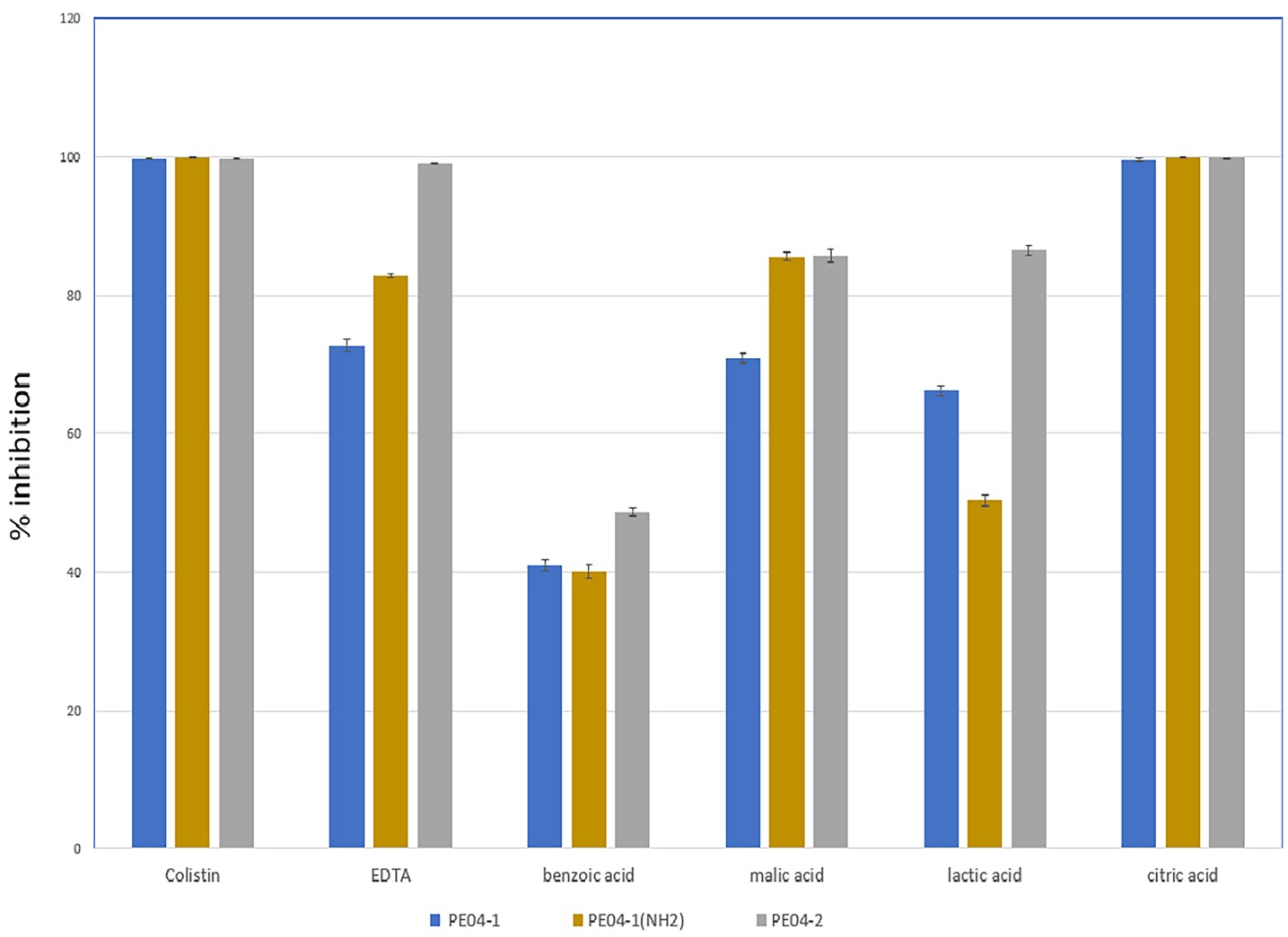

**Figure 2** **Screening for synergistic interaction of phage-encoded AMPs.** The growth inhibition rate (bar graph) of PE04-1 (blue), PE04-1 (NH$_2$) (brown), and PE04-2 (grey) combined with colistin, EDTA, and outer membrane permeabilizing agents (citric acid, lactic acid, malic acid, and benzoic acid) at 0.25× MIC of two agents. Data are expressed as the mean percentage ± SD of triplicate experiments.

## Biofilm eradication assay of phage-encoded peptides

We determined the peptides' ability to inhibit biofilm formation using the MIC values of the peptides. Percent inhibition of all three peptides showed that PE04-1, PE04-1(NH$_2$), and PE04-2 showed the ability to inhibit biofilm formation against twenty *A. baumannii* strains with the average percent inhibition as 58.11%, 75.51%, and 91.92%, respectively as shown in Figs. 4A, 4B and Table S6. Peptides PE04-2, a CPP-modified peptide, showed strong antibiofilm effects compared to the other peptides ($p < 0.05$) (Fig. 4B).

## *G. mellonella* infection assays of phage-encoded peptides

We determined peptide efficacy as a therapeutic approach in *G. mellonella* model. Larvae treated with PE04-1, PE04-1(NH$_2$), and PE04-2 at the MIC level showed survivability of XDR-AB infection of more than 80% at day 10. PE04-1(NH$_2$), and PE04-2 showed high

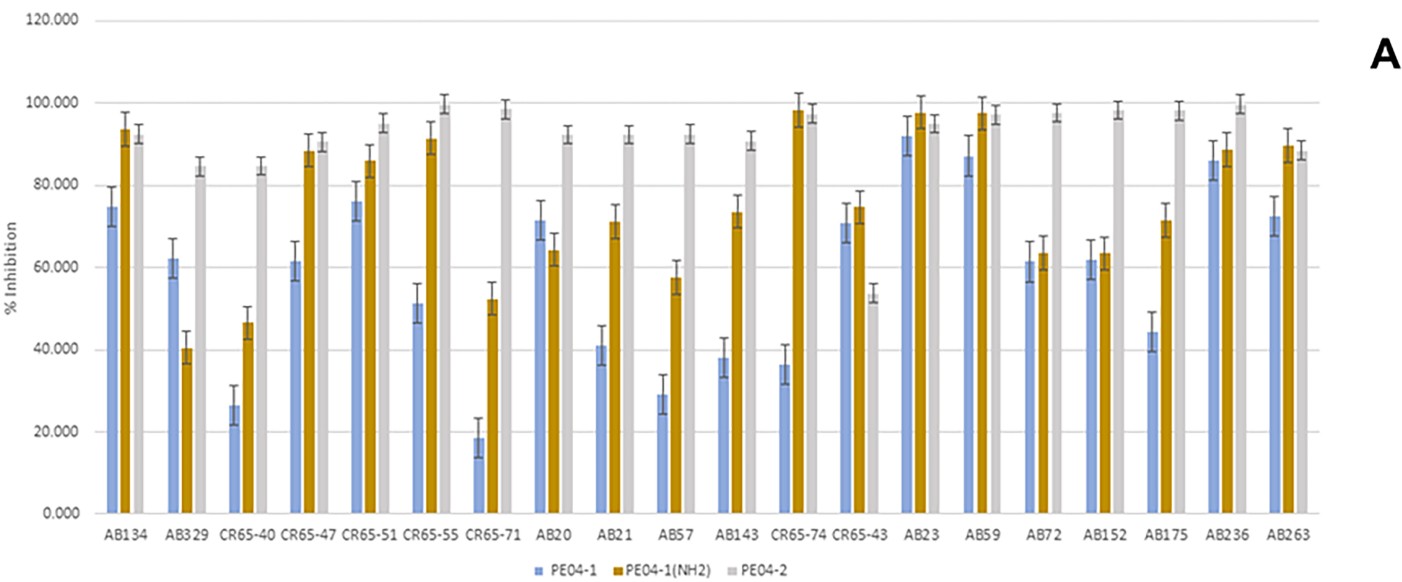

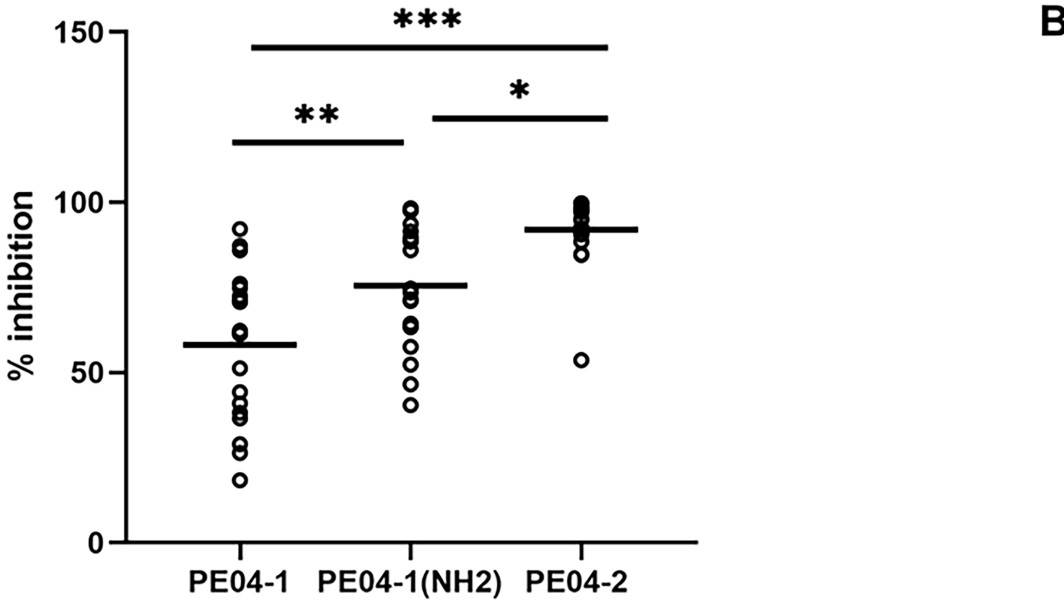

**Figure 3 Effect of phage-encoded AMPs on biofilm formation.** (A) Data represent the percentage of biofilm reduction in *A. baumannii* treated with PE04-1 (blue), PE04-1 (NH$_2$) (brown), and PE04-2 (grey) *vs.* control wells across 20 representative *A. baumannii* strains, including MDR-AB, XDR-AB, CR-AB, and colistin-resistant strains. Data are shown as the mean of three independent experiments ± SD. (B) Comparison of biofilm reduction ability among PE04-1, PE04-1 (NH$_2$), and PE04-2. Asterisks indicate statistical significance (*$p < 0.5$, **$p < 0.1$, ***$p < 0.01$).

survivability compared to PE04-1 (Fig. 5). Larvae injected with a dose of *A. baumannii* ATCC19606 showed a rapid decline in larvae number with less than 50% survival at day 5, while uninfected larvae injected with PBS or peptides, used as controls, a 100% survivability were obtained (Figs. S1–S3).

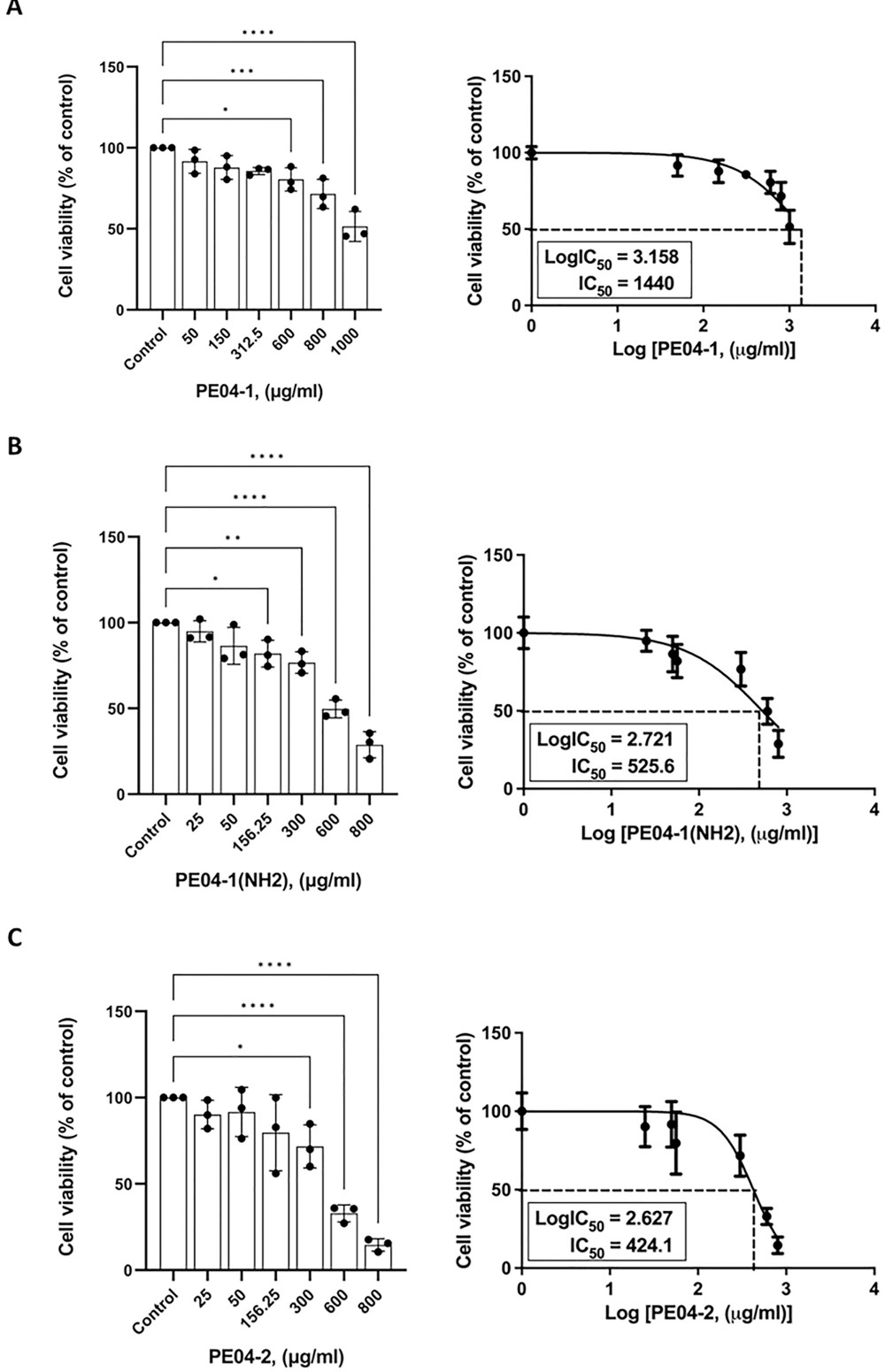

**Figure 4 Effects of peptides on cell viability and proliferation using the MTT assay.** Cells were treated with PE04-1 (A), PE04-1 (NH$_2$) (B), and PE04-2 (C) at MIC concentrations. Data are expressed as the mean of three independent experiments ± SD. Asterisks indicate statistical significance compared to the control (*$p < 0.5$, **$p < 0.1$, ****$p < 0.001$).

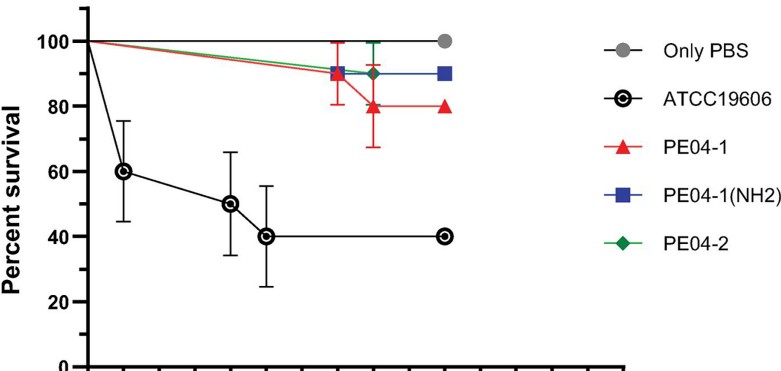

**G. mellonella infection assays of phage derived peptides**

**Figure 5 The survival rate of _Galleria mellonella_ larvae infected with extensively drug-resistant (XDR) _A. baumannii_ strains (AB-329) and treated with phage-encoded AMPs.** Survival rates of the larvae in different treatment groups were monitored for 10 days. The plotted points represent mortality events in PBS (control), AB329 (positive control), and AB329 mixed with peptides (PE04-1, PE04-1 (NH$_2$), PE04-2). All experimental groups were performed in triplicate.

## DISCUSSION

Antimicrobial peptides from bacteriophage-derived proteins are emerging as innovative tools in the fight against antimicrobial-resistant bacteria. Bacteriophage-derived proteins such as the tail-associated protein and endolysin of _A. baumannii_ bacteriophage have been shown to possess significant antibacterial properties, offering potential candidates for _in silico_ methods to identify antimicrobial peptides (_Lai et al., 2016_; _Sitthisak et al., 2023_; _Tu et al., 2023_). In addition, inhibitors of bacterial protein synthesis are promising lead compounds for developing new antimicrobials with a mechanism that interferes with bacterial DNA replication and bacterial protein synthesis pathways (_Mardirossian et al., 2020_; _Anandabaskar, 2021_). Thus, we analyzed the whole genome sequences of vPhT2 (MN864865) and vChT04 (OQ858591) and designed AMPs from tail-associated proteins, endolysins, and inhibitors of bacterial protein synthesis. We found only three peptides derived from endolysin, PE04-1, PE04-2, and PE04-1(NH$_2$), showed good activity (MIC ranging from 156.25–312.5 µg/ml). The endolysin from vPhT2 and vChT04 phage belongs to the lysozyme-like superfamily domain and showed the ability to cleave the β-(1,4)-glycosidic bond between N-acetylglucosamine and N-acetylmuramic acid, which are critical components of peptidoglycan found in bacterial cell walls (_Vermassen et al., 2019_).

Based on the MIC results, peptides derived from vChT04 appear to be the most promising peptides for further study, as they exhibited moderate to good antimicrobial activity against ATCC19606. Previous studies have investigated the use of phage-encoded antimicrobial peptides (AMPs) from endolysin against _A. baumannii_ that exhibited

antibacterial activity (*Thandar et al., 2016*; *Peng et al., 2017*; *Li et al., 2021*). We found that the peptides PE04-1, PE04-1(NH$_2$), and PE04-2 showed a potent and broad spectrum in killing MDR-AB, XDR-AB, and other Gram-negative bacteria. In addition, the peptides also showed antimicrobial activity against Gram-positive bacteria such as *S. aureus* and *B. subtilis*. The antibacterial activity of the PE04 peptide was determined by AntiBP2 and predicted that PE04 sequences showed similarity to the mammal's cathelicidin antimicrobial peptide (Table S1). Cathelicidins, a specific family of AMPs, are known for their broad-spectrum antimicrobial properties. Cathelicidins can disrupt microbial membranes through mechanisms such as pore formation, membrane thinning, and lipid segregation (*Agadi et al., 2022*). In this study, we proposed mechanisms of action of the AMPs in relation to membrane-targeting action. Compared to the other peptides, PE04-1, PE04-1(NH$_2$), and PE04-2 are cationic peptides with a positive charge (+5 to +6). They exhibited high hydrophobic ratios (48–52%) and high hydropathy values (0.07–0.38). Previous studies showed that the antimicrobial activity of AMPs is significantly influenced by their hydrophobicity and net charge due to the properties of the peptides to interact with microbial membranes and influence the peptides' ability to disrupt or penetrate these membranes (*Tan et al., 2021*; *Garvey, 2023*). In addition, peptide charge also serves as a critical determinant of antimicrobial activity, influencing interactions with microbial membranes and modulating selectivity and specificity (*Zhang et al., 2021*). Additionally, amidated peptides (PE04-1(NH$_2$)) and CPP-peptides (PE04-2) increase their antimicrobial activity compared to wildtype (PE04-1). Increased antimicrobial activity of these peptides can be explained by the positively charged amine group, and cationic CPPs peptide enhanced its ability to enter cells by interacting with the negatively charged bacterial membrane (*Tan et al., 2021*; *Garvey, 2023*).

This research study found a strong synergistic effect of three peptides with colistin and citric acid, which showed high inhibition percentages and low FIC indexes. In addition, the efficacy of peptide combinations with EDTA, malic acid, lactic acid, and benzoic acid are varied, with generally lower inhibition percentages compared to colistin and citric acid (Fig. 2). Colistin disrupts the bacterial outer membrane (OM) and cytoplasmic membrane (CM) by selectively targeting lipopolysaccharide (LPS) (*Sabnis et al., 2021*), potentially making bacteria more susceptible to peptide action. Citric acid is a weak acid with antimicrobial activity that can chelate divalent cations, which are essential for maintaining the structural integrity of LPS in the outer membrane and destabilize the LPS layer, making the bacteria more susceptible to antimicrobial peptides and other agents (*Burel, Kala & Purevdorj-Gage, 2021*). These findings highlight the potential for developing combination therapies using peptides and traditional antimicrobial agents to combat MDR-AB.

Biofilm assays of three peptides were determined. These observations are in agreement with previous reports demonstrating all peptides were able to inhibit biofilm formation in *A. baumannii* strains, indicating their potential as antibiofilm agents (*Peng et al., 2017*). However, the different peptides showed varying levels of inhibition, suggesting that each

peptide may have a unique mechanism of action or specific binding properties. Peptides PE04-2, a CPP-modified peptide, showed strong antibiofilm effects. The properties of PE04-2, which are cationic, amphipathic, and hydrophobic, make the peptide enter cells by interacting with the negatively charged biofilm extracellular matrix, which is composed of various components such as polysaccharides, proteins, and DNA (*Yasir, Willcox & Dutta, 2018*).

In order to apply the peptides as an antibacterial agent, we performed cytotoxicity assays of antimicrobial peptides to evaluate their potential toxicity toward human cells. Cytotoxicity assays of three peptides showed that all peptides exhibited no cytotoxicity when treated HepG2 cell lines at concentrations of up to MIC levels. In agreement with other studies, none of the phage peptides showed significant cytotoxicity toward human cell lines, ensuring their safety for potential therapeutic applications (*Lim et al., 2024*).

We used the *G. mellonella* infection model to evaluate the efficacy of phage peptides as a therapeutic approach against two strains of *A. baumannii*. The rapid decline in infected larvae with *A. baumannii* showed a significant decrease in survival, highlighting the pathogenicity of *A. baumannii* and its lethal impact on the larvae. Uninfected larvae injected with either PBS or the peptides (used as controls) maintained a 100% survival rate throughout the observation period. This indicates that the injection procedure and the peptides themselves are not inherently harmful to the larvae. Larvae treated with peptides PE04-1, PE04-1(NH$_2$), and PE04-2 at the MIC levels showed significant improvement in survival rates. By day 10, the survival rate exceeded 80% for all these treatments, with PE04-1(NH$_2$) and PE04-2 demonstrating particularly high efficacy compared to PE04-1. This study suggests that phage-encoded peptides, especially PE04-1(NH$_2$) and PE04-2, hold promise as effective therapeutic agents against *A. baumannii* infections in the *G. mellonella* model.

The current study has some limitations such as different peptides exhibited varying level of inhibition and may have unique mechanism of action. Future research could focus on specific mechanisms, such as specific binding properties, pore formation and membrane disruption of the peptides. The potential for resistance development of the peptides and the function of the peptides to modulate the immune response should be investigated. These will enhance the understanding and application of phage-encoded AMPs in antimicrobial therapies.

## CONCLUSIONS

This current study highlights the potent antimicrobial activity of phage-encoded AMPs, particularly PE04-1, PE04-1(NH$_2$), and PE04-2, against *A. baumannii*. Their effectiveness is significantly enhanced when used in combination with colistin and outer membrane permeabilizing agents. Additionally, these peptides reduce biofilm formation and improve survival rates in infection models. Thus, the peptides hold potential as therapeutic agents against MDR-AB, warranting further investigation for their possible therapeutic applications.

## ACKNOWLEDGEMENTS

The authors would like to thank the hospital's staffs for collecting the bacterial isolates. Language editing was performed using generative AI (ChatGPT and Grammarly).

### Funding

This work was supported by Naresuan University (NU), and National Science, Research and Innovation (NSRF) Grant No. R2566B044 to Sutthirat Sitthisak. The funders had no role in study design, data collection and analysis, decision to publish, or preparation of the manuscript.

### Grant Disclosures

The following grant information was disclosed by the authors:
Naresuan University (NU), and National Science, Research and Innovation (NSRF): R2566B044.

### Competing Interests

Sutthirat Sitthisak is an Academic Editor for PeerJ.

### Author Contributions

- Punnaphat Rothong performed the experiments, analyzed the data, prepared figures and/or tables, authored or reviewed drafts of the article, and approved the final draft.
- Udomluk Leungtongkam performed the experiments, prepared figures and/or tables, and approved the final draft.
- Supat Khongfak performed the experiments, analyzed the data, prepared figures and/or tables, and approved the final draft.
- Chanatinat Homkaew performed the experiments, prepared figures and/or tables, and approved the final draft.
- Sirorat Samathi performed the experiments, prepared figures and/or tables, and approved the final draft.
- Sarunporn Tandhavanant analyzed the data, prepared figures and/or tables, authored or reviewed drafts of the article, and approved the final draft.
- Jatuporn Ngoenkam conceived and designed the experiments, performed the experiments, analyzed the data, prepared figures and/or tables, and approved the final draft.
- Apichat Vitta conceived and designed the experiments, performed the experiments, authored or reviewed drafts of the article, and approved the final draft.
- Aunchalee Thanwisai conceived and designed the experiments, performed the experiments, authored or reviewed drafts of the article, and approved the final draft.
- Sutthirat Sitthisak conceived and designed the experiments, performed the experiments, analyzed the data, prepared figures and/or tables, authored or reviewed drafts of the article, and approved the final draft.

## Ethics

The following information was supplied relating to ethical approvals (*i.e.*, approving body and any reference numbers):

The protocol for handle bacterial pathogens was approved by the Naresaun University Institutional Biosafety Committee and Naresuan University Animal Care and Use Committee (protocol No. NU-AI660602).

## DNA Deposition

The following information was supplied regarding the deposition of DNA sequences:

The sequences are available at Genbank: QHJ75684.1, QHJ75684.1, QHJ75726.1, QHJ75785.1, QHJ75776.1, WIS40123.1, WIS40082.1, WIS40123.1, WIS40123.1.

## Data Availability

The raw data is available in the Supplemental Files.

## Supplemental Information

Supplemental information for this article can be found online at http://dx.doi.org/10.7717/peerj.18722#supplemental-information.

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
