# Peer review of "Antimicrobial activity and synergistic effect of phage-encoded antimicrobial peptides with colistin and outer membrane permeabilizing agents against Acinetobacter baumannii"

_PeerJ, doi:10.7717/peerj.18722_

## Round 0.1 · original submission · Major Revisions

Please address concerns of all reviewers and amend your manuscript accordingly.

Reviewer 1 ·

Basic reporting

The manuscript by Punnaphat Rothong and colleagues presents the findings of their investigation into phage-encoded antimicrobial peptides (AMPs) and their influence on infections caused by Acinetobacter baumannii. This study builds upon the authors' previous research regarding the application of the phage vB_AbaM_PhT2 endolysin. The authors evaluated several AMPs derived from endolysin, tail lysozyme, and tail fibers (possibly tail spikes) of phages vB_AbaAut_ChT04 and PhT2. Their comprehensive research included experiments utilizing the Galleria mellonella model, revealing that several AMPs demonstrate potential as effective therapeutic agents against A. baumannii infections. Additionally, the study examined the peptides' ability to inhibit biofilm formation and highlighted the synergistic effects of the engineered AMPs when combined with colistin and citric acid, achieving significant inhibition percentages and low fractional inhibitory concentration indices.

Overall, the manuscript is well-structured, and the methodologies employed appear to be sound. The illustrations are of high quality and effectively convey the results. However, a significant concern arises from the lack of characterization of the phages utilized in the study. It is essential to provide a short description of these phages to clarify the rationale behind selecting these specific proteins. Do these phages belong to the Autographiviridae family? Furthermore, there is a notable absence of analysis and interpretation regarding the mechanisms of action of the AMPs, which needs some explanation.

Specific comments:

Line 78: Are you referring to the enzymatic domains of the tail spikes? Please clarify the predicted functions of the cell-wall degrading proteins used for the generation of the AMPs.

Lines 79-80: Have you considered using AlphaFold to model the structures and delineate the boundaries of the domains?

Experimental design

Experimental design is fine.

Validity of the findings

I didn't notice anything that inspired mistrust.

Reviewer 2 ·

Basic reporting

Rothong et al. evaluated three peptides PE04-1, PE04-1(NH2), and PE04-2 in terms of antibacterial activity against Gram-positive and Gram-negative bacteria.

Experimental design

My two major concerns consider, first biofilm formation assays, and second Galleria mellonella in vivo experiments.
It is easy to perform the biofilm formation assays, as the peptides can kill planktonic bacteria in the solution not allowing them to form biofilm. It says nothing about mature biofilm, and how the peptides act against 24 or 72 h formed biofilm. Therefore, the manuscript would benefit from performing such experiments.
Second, I am a bit surprised that in Galleria mellonella infection assay bacteria were mixed with peptides before injecting them to the larvae. Generally, even in the publication of Peleg et al, there are two options: pre-treatment or post-treatment. The first is to protect the larvae against infection and the second is to check the antimicrobial activity of selected agents or antibiotics in infected G. mellonella. I suggest the authors to conduct similar experiments.

Validity of the findings

What are the exact concentrations of colistin, EDTA, citric acid etc. in inhibition assay for synergy testing, 0.25 x MIC sounds enigmatic. The information should be provided in the main text.

Additional comments

In addition, the discussion requires some polishing in terms of language.

·

Basic reporting

Add keywords in the abstract
The time frame should be specified in the working method of the abstract
In the working method, the abstract of the statistical tests and the software used should be mentioned
In the abstract, the discussion and conclusion should be together and the discussion should be explained more

Experimental design

Instead of our work, the current research should be mentioned and research suggestions should be made
The statement of the problem is brief
In the statement of the problem, the topic should be explained more and the antimicrobial and permeable activity of the outer membrane should be discussed more.
Refer to similar studies
The reference Vaara et al., 1992 is very old
The purpose of the research should be stated at the end of the problem statement
Reference Lorian Ve. 2005 to be completed
The research ethics code should be mentioned
The data collection of 2015 has been completed. It seems that if the sample was collected in recent years, different results would have been obtained due to the new antibiotic resistance.
The procedure is correctly mentioned

Validity of the findings

The limitations of the research will be discussed at the end
Instead of Our study, the current study should be mentioned
In the conclusion, the suggestions of the research should be made

Additional comments

Regarding clear, unambiguous and professional English, please check
The number of references should be increased according to the subject of the study
At least 40 references

---

## Round 0.2 · Minor Revisions

Please address the remaining concern of the reviewer #2 and amend text accordingly.

Reviewer 1 ·

Basic reporting

Everything is fine.

Experimental design

Everything is fine.

Validity of the findings

Everything is fine.

Additional comments

The authors have addressed all comments thoroughly and constructively. The revisions significantly enhance the manuscript’s scientific rigor and clarity. The high quality of the research and presentation warrants publication in PeerJ.

Reviewer 2 ·

Basic reporting

Although, the authors did not fully addressed my queries, after reading of the re-submitted manuscript I think it is significantly improved. Only one concern regards the concentration of peptides in MTT assay. In lines 296 – 297, the IC50 of peptides PE04-1, PE04-1(NH2) and PE04-2 are given in mg/ml (1,440, 525.6 and 424.1 mg/ml). While in the previous version the units were µg/ml, is it a mistake? The same applies to line 291.

Experimental design

Experimental design is appropriate.

Validity of the findings

The results presented in the manuscript will be interesting to the readers of PeerJ.

·

Basic reporting

The requested amendments have been implemented and are acceptable.

Experimental design

The requested amendments have been implemented and are acceptable.

Validity of the findings

The requested amendments have been implemented and are acceptable.

Additional comments

With regards
Thanks for the advice
The requested amendments have been made and are acceptable.

---

## Round 0.3 · accepted · Accept

All remaining issues were addressed and revised manuscript is acceptable now.